# Highly Effective Proton-Conduction Matrix-Mixed Membrane Derived from an -SO_3_H Functionalized Polyamide

**DOI:** 10.3390/molecules27134110

**Published:** 2022-06-26

**Authors:** Jamal Afzal, Yaomei Fu, Tian-Xiang Luan, Zhongmin Su, Pei-Zhou Li

**Affiliations:** 1School of Chemistry and Chemical Engineering, Shandong University, No. 27 Shanda South Road, Jinan 250100, China; jamalsarwari46@gmail.com (J.A.); txluan@mail.sdu.edu.cn (T.-X.L.); 2Shandong Peninsula Engineering Research Center of Comprehensive Brine Utilization, Weifang University of Science and Technology, Shouguang 262700, China; fuyaomei@wfust.edu.cn (Y.F.); zmsu@nenu.edu.cn (Z.S.); 3School of Chemistry and Environmental Engineering, Changchun University of Science and Technology, Changchun 130022, China; 4Science Center for Material Creation and Energy Conversion, Institute of Frontier and Interdisciplinary Science, Shandong University, Qingdao 266237, China

**Keywords:** fuel cell, polyamide, matrix-mixed membrane, proton conduction, sulfonic acid

## Abstract

Developing a low-cost and effective proton-conductive electrolyte to meet the requirements of the large-scale manufacturing of proton exchange membrane (PEM) fuel cells is of great significance in progressing towards the upcoming “hydrogen economy” society. Herein, utilizing the one-pot acylation polymeric combination of acyl chloride and amine precursors, a polyamide with in-built -SO_3_H moieties (**PA-PhSO_3_H**) was facilely synthesized. Characterization shows that it possesses a porous feature and a high stability at the practical operating conditions of PEM fuel cells. Investigations of electrochemical impedance spectroscopy (EIS) measurements revealed that the fabricated **PA-PhSO_3_H** displays a proton conductivity of up to 8.85 × 10^−2^ S·cm^−1^ at 353 K under 98% relative humidity (RH), which is more than two orders of magnitude higher than that of its -SO_3_H-free analogue, **PA-Ph** (6.30 × 10^−4^ S·cm^−1^), under the same conditions. Therefore, matrix-mixed membranes were fabricated by mixing with polyacrylonitrile (PAN) in different ratios, and the EIS analyses revealed that its proton conductivity can reach up to 4.90 × 10^−2^ S·cm^−1^ at 353 K and a 98% relative humidity (RH) when the weight ratio of **PA-PhSO_3_H**:PAN is 3:1 (labeled as **PA-PhSO_3_H-PAN (3:1)**), the value of which is even comparable with those of commercial-available electrolytes being used in PEM fuel cells. Additionally, continuous tests showed that **PA-PhSO_3_H-PAN (3:1)** possesses a long-life reusability. This work demonstrates, using the simple acylation reaction with the sulfonated module as precursor, that low-cost and highly effective proton-conductive electrolytes for PEM fuel cells can be facilely achieved.

## 1. Introduction

Energy depletion and environmental degradation are the two most pressing issues confronting humanity in the current century. As the consumption of fossil energy grows along with the global population, it is unable to meet the rising energy demands [1,2]. Meanwhile, the speed of searching for renewable energy sources is limited by economic regulations, natural laws, and societal acceptability. Economic standards and cultural acceptability are malleable in comparison with natural principles, which must be understood and investigated at the molecular and atomic levels to implant and discover new alternative energy sources. Renewable and innovative clean energy sources are being developed, and significant efforts have been made in this sector during the previous few decades [3,4,5,6]. Solar energy, hydroelectric energy, nuclear energy, and geothermal energy are some of the emerging renewable energy sources. Although hydrogen is not an inherent energy source in nature, it is regarded as the best energy carrier for resolving energy- and environment-related issues owing to their high energy density and low carbon emission [3,4,5,6]. Therefore, hydrogen-based fuel cells, notably proton-exchange membrane (PEM) fuel cells, have gradually become the most effective alternative technology to replace engine technology based on traditional fossil fuels [7,8,9]. The proton exchange membrane is a crucial component of PEM fuel cells, as it highly affects the performance of the whole integrated fuel cell. As a result, developing a low-cost and effective proton-conductive electrolyte to overcome the issues standing in the way of large-scale manufacturing of proton exchange membranes and fuel cells becomes a significantly important mission on the path to the upcoming “hydrogen economy” society.

Great efforts have been dedicated to this mission, and numerous materials such as polyoxometalates [10,11], graphynes [12,13], recently developed porous materials [14,15,16,17], and so on have been constructed and studied thus far as proton-conductive electrolytes for application in PEM fuel cells. Among these functional materials, Nafion, a perfluorinated sulfonated polymer that shows an excellent proton conductivity and durability has gradually become a commercially accessible PEM electrolyte, although many issues in its fabrication and application still need to be overcome [18,19,20]. Following this accomplishment, other materials containing sulfonated (-SO_3_H) functional groups have been produced and extensively studied as PEM candidates to be used in PEM fuel cells [21,22,23,24,25,26,27]. Recently, using the Cu(I)-CAAC click reaction [28,29,30,31,32], we also inserted the -SO_3_H groups into the skeleton of a triazole-based porous organic polymer (TaPOP-1) and a robust metal organic framework (MOF), UiO-66, by the incorporation of -SO_3_H groups during the in-situ synthesis process or post-modification treatment, respectively. The obtained sulfonated samples of TaPOP-1-SO_3_H [28] and UiO-66-SO_3_H [29] both exhibit remarkable enhancements in proton conductivity and long-life reusability.

Polyamides are a group of functional organic polymers which have been large-scale manufactured with mature processes in the chemical industry for many daily necessities such as nylon [33,34,35,36]. After proton-conductive modification, it should be a group of high potential proton-conductive electrolytes. Thus, herein, employing the facile one-pot polymeric acylation of acyl chloride and amine precursors, we report the successful fabrication of a polyamide with in-built -SO_3_H moieties, designated as **PA-PhSO_3_H**, for use as a proton-conductive electrolyte in PEM fuel cells (Figure 1). Characterization shows that it possesses a porous feature and a high stability at the practical operating conditions of PEM fuel cells. Proton conductivity studies revealed that it had a value of 8.85 × 10^−2^ S·cm^−1^ at 353 K and a 98% relative humidity (RH), which is more than two orders of magnitude higher than that of its -SO_3_H-free analogue, **PA-Ph** (6.30 × 10^−4^ S·cm^−1^), under the same conditions. When it was incorporated into matrix-mixed membranes with varying amounts of polyacrylonitrile (PAN), similar values to those of commercially available proton-conductive electrolytes were observed under practical operating conditions. Considering the mature processes of polyamides in the chemical industry, this work successfully demonstrates that by incorporation of -SO_3_H functionalities into the skeletons of low-cost polyamides, highly effective proton-conductive electrolytes for application in PEM fuel cells can be produced.

## 2. Results and Discussion

To synthesize proton-conductive electrolytes with inexpensive reactants, a -SO_3_H functionalized phenylenediamine [21,22,23,24] and a triple-symmetrical 1,3,5-benzenetricarbonylchloride were taken as precursors. Following the simple one-pot polymeric acylation process and centrifugation collection method, basic procedures such as thorough washing and natural drying were used to yield a fine grey powder of the synthesized **PA-PhSO_3_H** (Figure 1) [33,34,35,36]. As a control experiment, a similar procedure was used to synthesis the -SO_3_H-free analogue, **PA-Ph**.

### 2.1. Morphology and Structural Analyses

Unsurprisingly, the powder X-ray diffraction analysis of the resultant fine grey powder revealed an amorphous morphology for both samples (Appendix A). Fourier-transform infrared spectroscopy was used to analyze the formation of amide linkages in the synthesized polyamides, **PA-PhSO_3_H** and **PA-Ph**. As illustrated in Figure 2a and Appendix A, sharp peaks at 1185 and 1244 cm^−1^ were observed in the synthesized **PA-PhSO_3_H** and **PA-Ph**, respectively, which can be attributed to the -C-N- stretching vibration of the amide group formed in-situ during the polymeric acylation reaction processes, confirming the successful formation of the amide linkage in the synthesized samples [33,34,35,36]. The strong absorption band at 1025 cm^−1^, which corresponds to the characteristic peak of -SO_3_H moieties [21,27], was also identified in **PA-PhSO_3_H**, indicating that the sulfonated functional groups were also effectively integrated into the skeleton of the synthesized **PA-PhSO_3_H**. Scanning electron microscopy (SEM) images were subsequently analyzed to observe the morphology of the synthesized **PA-PhSO_3_H**, and a fine amorphous powder was observed in the resultant images as shown in Appendix A, which agrees very well with the results obtained from the PXRD measurements. Energy-dispersive X-ray (EDX) spectroscopy mapping investigations were also performed during the SEM observations, revealing that the elements C, N, O, and S are evenly dispersed in the synthesized **PA-PhSO_3_H** (Figure 2b), further demonstrating the successful incorporation of the -SO_3_H functional groups into the polyamide.

### 2.2. Porosity and Thermal Stability Analyses

N_2_ sorption measurements were also carried out at 77 K, and the results revealed that both the prepared samples had a porous feature with surface areas of 36 and 51 m^2^·g^−1^ for **PA-PhSO_3_H** and **PA-Ph**, respectively (Figure 3a) [37]. The thermal stability of both samples was then determined using thermogravimetric analyses. When the temperature increased from room temperature up to 150 °C, weight loss was observed in both samples, which should be attributed to the loss of absorbed water molecules in their pores [27]. When the temperature was approximately 250 °C, significant weight loss due to polymer decomposition was observed in both samples (Figure 3b), indicating that they exhibit excellent thermal stability at the operating temperature of a PEM fuel cell (typically lower than 120 °C) [7,8,9].

### 2.3. Impedance Analysis and Proton Conductivity of the Polyamides

After pressing the synthesized samples of both polyamides, **PA-PhSO_3_H** and **PA-Ph**, into tablets, proton conductivity was measured using electrochemical impedance spectroscopy (EIS) at temperatures ranging from 303 K to 353 K and under various relative humidity conditions. As shown in Figure 4 and Table 1, the results of EIS measurement of both materials demonstrated that their proton conductivity values rise as the temperature and relative humidity increase. When the operating temperature is increased to 80 °C and the relative humidity is 98% RH, the proton conductivity of **PA-PhSO_3_H** rises up to 8.85 × 10^−2^ S·cm^−1^, which is more than two orders of magnitude higher than that of its -SO_3_H free analogue **PA-Ph**, which has a value of only 6.30 × 10^−4^ S·cm^−1^ under the same conditions. The values of proton conductivity of **PA-PhSO_3_H** are comparable to those of commercially-available proton-conductive electrolytes evaluated under the same conditions [18,19,20]. After calculating Arrhenius plots for proton conductivity, the activation energy values were then computed using the least-squares fitting method. The activation energies of **PA-PhSO_3_H** were 0.31, 0.30, and 0.24 eV at 98%, 85%, and 75% RH, respectively (Figure 4d). These results imply that the -SO_3_H functionalized polyamide synthesized in this work follows the Grotthuss mechanism [38,39,40,41]. The amide linkages in the synthesized **PA-PhSO_3_H** usually have an affinity interaction with the adsorbed water molecules and the in-built -SO_3_H moieties can offer very active free-moving protons to improve the activity of proton conduction. Thus, the high performance of proton conductivity of the fabricated **PA-PhSO_3_H** should be ascribed to the synergistic impact of the in-situ formed amide groups and the in-built -SO_3_H moieties, which make it a promising candidate as an efficient proton-conductive electrolyte [21,22,23,24].

### 2.4. Matrix-Mixed Membrane Fabrication and Characterizations

In response to these encouraging results of proton conduction, membrane fabrication was then carried out using **PA-PhSO_3_H** as the core proton-conductive electrolyte. Matrix-mixed membranes were fabricated by uniformly mixing **PA-PhSO_3_H** with polyacrylonitrile (PAN) in various ratios, labeled as **PA-PhSO_3_H-PAN (n:m)** (n:m represents the weight ratios of **PA-PhSO_3_H**:PAN), and painting the evenly mixed viscous material on the surface of aluminum foil, allowing it to dry naturally, and soaking it in water (Figure 5a) [42]. When the ratio of **PA-PhSO_3_H** to PAN was more than 4:1, the fabricated matrix-mixed membranes became fragile. Thus, matrix-mixed membranes with a maximum ratio of **PA-PhSO_3_H** and PAN of 3:1 (labeled as **PA-PhSO_3_H-PAN (3:1)**) were fabricated and investigated. SEM measurements were then subsequently used to inspect the morphology of the fabricated membranes. As shown in Figure 5b, the SEM image of the aerial view of the representative **PA-PhSO_3_H-PAN (3:1)** revealed a very flat and perfect surface, while its side-view image revealed it had a thickness of about 96.4 μm which agrees well with the thickness of ~100 nm measured by vernier caliper.

The physicochemical properties of our prepared membranes were then investigated according to the reported standard methods [43,44,45,46]. Firstly, an acid–base titration method was exploited to determine the ion exchange capacity (IEC) of the fabricated matrix-mixed membranes [43,44]. After immersing the pre-weighed dried membranes in a 2 M NaCl solution for one day to replace the protons with sodium ions, the solution was titrated using a 0.01 M NaOH solution (see details in Appendix A). The calculation results showed that the IEC of the fabricated matrix-mixed membranes increases from 1.57 mmol/g for **PA-PhSO_3_H-PAN (0.1:1)** to 2.89 mmol/g for **PA-PhSO_3_H-PAN (3:1)**, which agrees well with their theoretical values as shown in Table 2. Then, the water uptake properties and swelling characteristics, as well as the dimensional stability, were also tested for the fabricated matrix-mixed membranes (see details in Appendix A) [45,46]. After immersing the pre-weighed dried membranes overnight in distilled H_2_O at 30 and 80 °C, respectively, they were then rapidly dried by filter paper and weighed to calculate the water uptake. Moreover, the results showed that all the membranes show higher water uptake capabilities when treated with higher temperatures, and the water uptake capabilities increase along with the sample ratio increase as shown in Table 2. With the same method, swelling characteristics and dimensional stability were also calculated. All of the fabricated matrix-mixed membranes show a high dimensional stability with a swelling ratio of less than 7% as shown in Table 2.

Subsequently, the chemical and hydrolytic stabilities of the fabricated matrix-mixed membranes were also studied (see details in Appendix A). After immersing the prepared membranes in Fenton reagent (FeSO_4_ in H_2_O_2_ (3%), 3.0 ppm) at 30 °C and 80 °C, respectively, the oxidative chemical stability was determined by the elapsed time (t) it took for the membrane to dissolve completely [45], which shows that all of the membranes have an elapsed time (t) of more than 6 h at 80 °C and even longer when tested at 30 °C (Table 2). When they were immersed in distilled water at 50 °C, their elapsed time values until the membranes lost their mechanical properties were also recorded [44]. As shown in Table 2, all of the membranes show an elapsed time longer than 84 h, and the membrane of **PA-PhSO_3_H-PAN (3:1)** even shows an elapsed time of up to 120 h. All these tests revealed the fabricated matrix-mixed membranes possess good physicochemical properties and should be highly promising candidates for proton conductivity membranes applied in PEM fuel cells.

The high proton conductivity of the synthesized **PA-PhSO_3_H** and the successful fabrication of robust matrix-mixed membranes based on **PA-PhSO_3_H** motivated us to evaluate the proton conductivity of the fabricated membranes. The EIS analysis results revealed that the proton conductivity increases along with the sample ratio increase in the membranes; it is only 7.61 × 10^−3^ S·cm^−1^ for the membrane of **PA-PhSO_3_H-PAN (0.1:1)** at 353 K and 98% RH, while it reaches up to 4.90 × 10^−2^ S·cm^−1^ for the membrane of **PA-PhSO_3_H-PAN (3:1)** under the same conditions (Table 3 and Figure 6 and Appendix A), the value of which is even comparable with those of commercial-available electrolytes being used in PEM fuel cells at the same temperature and relative humidity [18,19,20]. Additionally, the continuous tests showed that all of the fabricated membranes possess long-life reusability (Figure 6 and Appendix A). As highly promising PEM candidates, we will test them in an integration PEM fuel cell in future works. This study reveals that by using the facile polymeric acylation reaction with the -SO_3_H module as a precursor, extremely efficient and robust proton-conductive electrolytes for PEM fuel cells can be achieved.

## 3. Materials and Methods

### 3.1. Materials

All the chemicals, reagents and solvents used for the materials’ syntheses were available commercially and were used as received unless specifically mentioned.

### 3.2. Synthesis of **PA-PhSO_3_H**

In a two-neck round-bottom flask (100 mL), 564 mg p-phenylenediamine sulfonic acid (3 mmol), 995 mg potassium carbonate (7.2 mmol), and 30 mL of anhydrous 1,4-dioxane were mixed. After stirring for 20 min, 531 mg 1,3,5-benzene tricarbonyl trichloride (2 mmol) in 20 mL anhydrous 1,4-dioxane was also added into the flask dropwise. Thus, after stirring and refluxing at 100 °C for 72 h in the presence of a nitrogen atmosphere, a solid powder was generated. After collection by centrifugation and washing with 0.1 M H_2_SO_4_ aqueous solution and absolute ethanol, and then acetone three times, respectively, and drying overnight in a vacuum oven at 75 °C, a fine grey powder of **PA-PhSO_3_H** was finally obtained (Figure 2) with a yield of 85% based on the starting reactants.

### 3.3. Synthesis of **PA-Ph**

The synthesis procedure was similar as that of **PA-PhSO_3_H**. Potassium carbonate (7.2 mmol), 1,4-diaminobenzene (3 mmol, 324.24 mg), and 30 mL of anhydrous 1,4-dioxane were added into a two-neck round-bottom flask, followed by the drip addition of an anhydrous 1,4-dioxane solution of 1,3,5-benzene tricarbonyl trichloride (2 mmol, 531 mg) under stirring and an N_2_ atmosphere. After refluxing for 72 h at 100 °C, the produced powder was collected by centrifugation and washing with 0.1 M H_2_SO_4_ aqueous solution and absolute ethanol, and then acetone three times, respectively. After drying overnight in a vacuum oven at 75 °C, a brownish powder of **PA-Ph** was finally obtained.

### 3.4. Fabrication of Matrix-Mixed Membranes

Matrix-mixed membranes were fabricated by phase inversion via immersion precipitation of the fabricated samples. Polyacrylonitrile (PAN) was used to form casting solutions as a polymer matrix. PAN was dissolved in DMF and put in an oven overnight at 70 °C to obtain a clear slurry. Then the samples of fabricated **PA-PhSO_3_H** and **PA-Ph** were mixed with the clear slurry, respectively, in different ratios to obtain a homogeneous solution. The solution was then placed across the aluminum foil’s surface with a 100 µm gap casting knife to give the formation of a coating film. After drying naturally and being immersed in water for one day to remove the additives and solvents, matrix-mixed membranes were successfully obtained.

## 4. Conclusions

In summary, to construct low-cost and effective proton-conductive electrolytes for application in PEM fuel cells, a polyamide with in-built -SO_3_H moieties was successfully synthesized by utilizing the facile one-pot polymeric acylation reaction. Characterization shows that it possesses a porous feature and a high stability at the practical operating conditions of PEM fuel cells. Proton conductivity studies reveal that it had a value of 8.85 × 10^−2^ S·cm^−1^ at 353 K under 98% RH, which is more than two orders of magnitude higher than that of its -SO_3_H-free analogue under the same conditions. When it was incorporated into matrix-mixed membranes with varying ratios of polyacrylonitrile (PAN), a value of up to 4.90 × 10^−2^ S·cm^−1^ was observed at 353 K and 98% RH, which is comparable to those of commercially-available proton-conductive electrolytes under the same conditions. Such a result makes the facile one-pot polymeric acylation a promising method to fabricate effective proton-conductive electrolytes by in-situ incorporation of -SO_3_H moieties into the skeletons of polyamides. This work sheds light on the facile fabrication of low-cost proton-conductive membranes for application in PEM fuel cells.

## Figures and Tables

**Figure 1 molecules-27-04110-f001:**
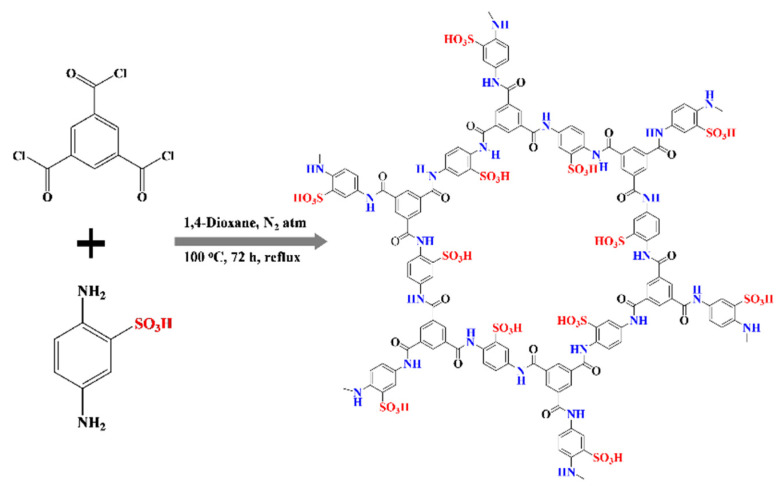
Synthetic scheme of **PA-PhSO_3_H**.

**Figure 2 molecules-27-04110-f002:**
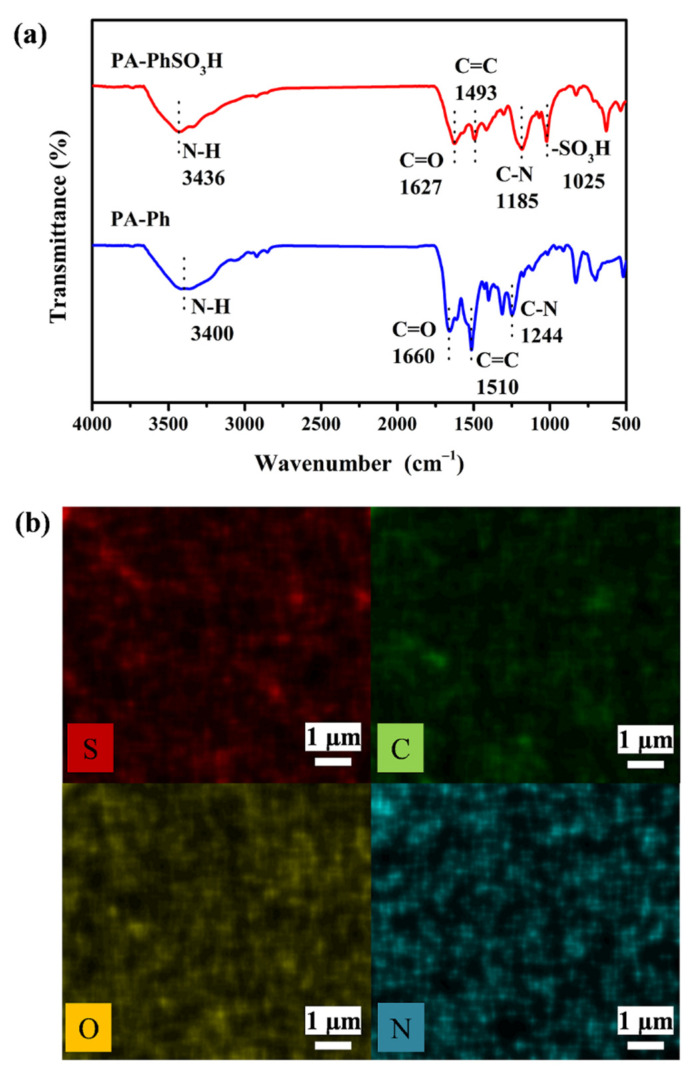
(**a**) FTIR spectra of **PA-PhSO_3_H** and **PA-Ph**. (**b**) EDX mapping images of **PA-PhSO_3_H**.

**Figure 3 molecules-27-04110-f003:**
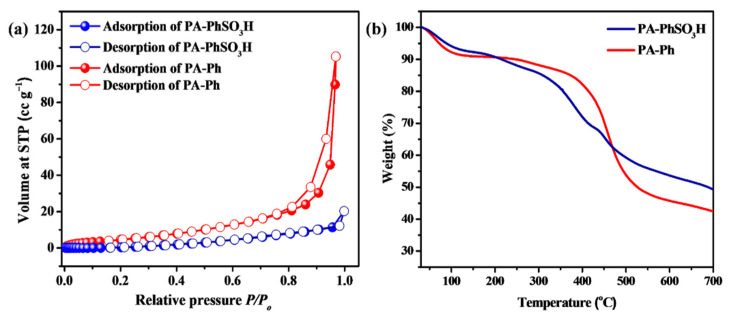
(**a**) N_2_ sorption isothermal and (**b**) TGA curves of **PA-PhSO_3_H** and **PA-Ph**.

**Figure 4 molecules-27-04110-f004:**
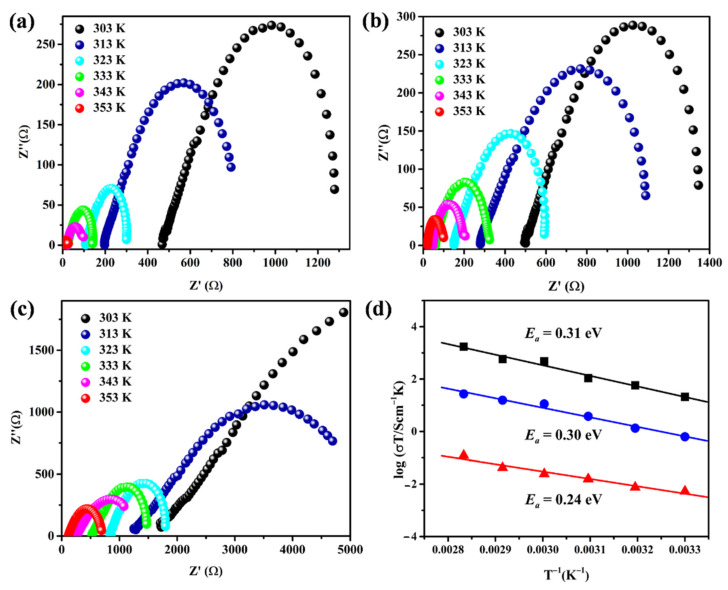
Nyquist plots of **PA-PhSO_3_H** measured at different temperatures and different relative humidity: (**a**) 98% RH, (**b**) 85% RH, (**c**) 75% RH. (**d**) Arrhenius plots of proton conductivity for **PA-PhSO_3_H** at 98% RH (black), 85% RH (blue), and 75% RH (red).

**Figure 5 molecules-27-04110-f005:**
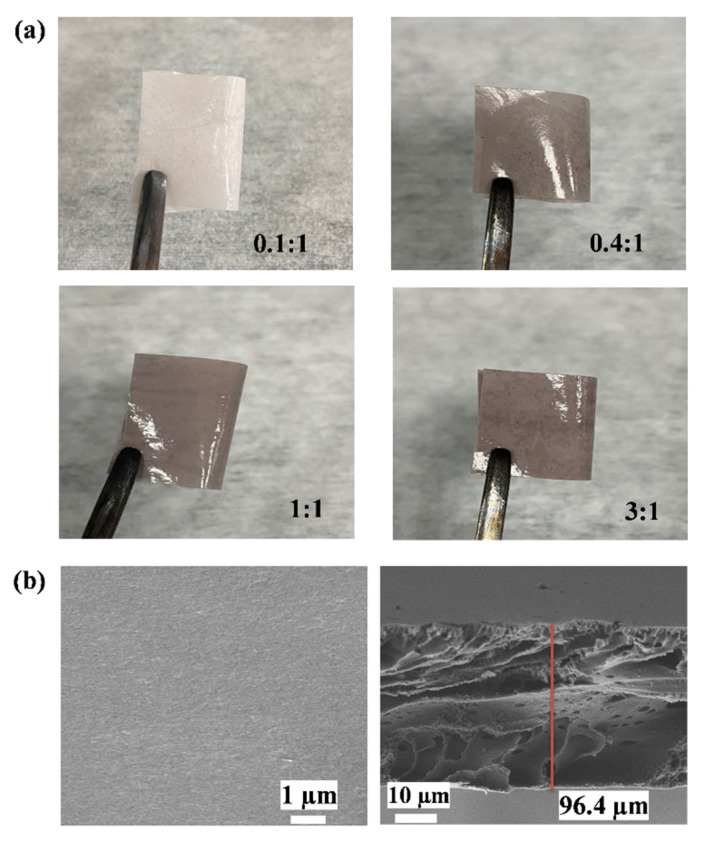
(**a**) Fabricated matrix-mixed membranes with different ratios of **PA-PhSO_3_H**:PAN. (**b**) SEM images of the upper surface (**left**) and thickness (**right**) of the fabricated matrix-mixed membrane **PA-PhSO_3_H-PAN (3:1)**.

**Figure 6 molecules-27-04110-f006:**
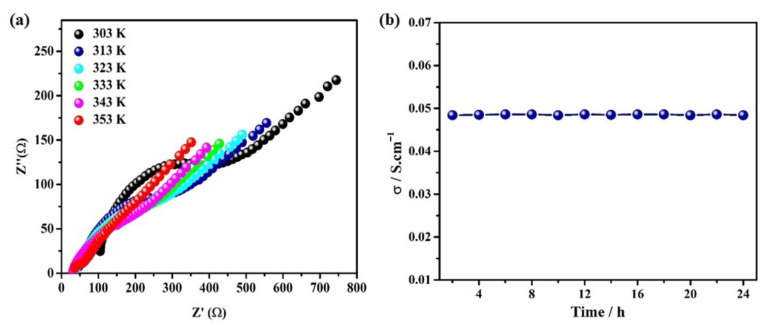
(**a**) Nyquist plot at 98%RH and different temperatures and (**b**) Long-life reusability test at 80 °C and 98% RH of **PA-PhSO_3_H-PAN (3:1)**.

**Table 1 molecules-27-04110-t001:** Temperature-dependent proton conductivities (S·cm^−1^) of the synthesized **PA-PhSO_3_H** under different relative humidity conditions.

Temperature	98%RH	85%RH	75%RH	63%RH	53%RH	43%RH	33%RH
80 °C	8.85 × 10^−2^	7.63 × 10^−2^	1.09 × 10^−2^	3.64 × 10^−4^	5.76 × 10^−6^	3.18 × 10^−6^	2.60 × 10^−6^
70 °C	5.32 × 10^−2^	4.55 × 10^−2^	3.86 × 10^−3^	1.30 × 10^−4^	3.66 × 10^−6^	2.69 × 10^−6^	2.23 × 10^−6^
60 °C	4.50 × 10^−2^	3.37 × 10^−2^	2.32 × 10^−3^	8.42 × 10^−6^	3.32 × 10^−6^	2.10 × 10^−6^	2.02 × 10^−6^
50 °C	1.17 × 10^−2^	1.05 × 10^−2^	1.54 × 10^−3^	6.48 × 10^−6^	2.13 × 10^−6^	1.86 × 10^−6^	1.80 × 10^−6^
40 °C	5.83 × 10^−3^	4.27 × 10^−3^	7.66 × 10^−4^	4.68 × 10^−6^	1.83 × 10^−6^	1.70 × 10^−6^	1.64 × 10^−6^
30 °C	2.18 × 10^−3^	2.07 × 10^−3^	5.58 × 10^−4^	3.31 × 10^−6^	1.68 × 10^−6^	1.55 × 10^−6^	1.17 × 10^−6^

**Table 2 molecules-27-04110-t002:** Ion exchange capacity (IEC), water uptake (WU), dimensional stability, swelling ratios, and chemical stability and hydrolytic stability of PA-PhSO_3_H-PAN with different ratios.

Membranes	IEC (mmol/g)		WU (%)		Dimensional Stability (%)				Swelling Ratio (%)		Chemical Stability		Hydrolytic Stability
					$\left(\Delta L_{c}\right)$		$\left(\Delta W_{c}\right)$				T (h)		T (h)
	IEC_T_	IEC_Exp._	30 °C	80 °C	30 °C	80 °C	30 °C	80 °C	30 °C	80 °C	30 °C	80 °C	50 °C
**PA-PhSO_3_H-PAN (3:1)**	2.95	2.89	16.22	28.13	3.32	4.21	4.08	5.62	4.66	6.80	18	18	120
**PA-PhSO_3_H-PAN (1:1)**	2.60	2.52	8.14	15.33	2.54	2.87	2.14	4.46	2.54	3.89	18	18	120
**PA-PhSO_3_H-PAN (0.4:1)**	1.91	1.85	4.76	9.20	0.95	1.18	1.24	1.98	1.47	3.03	18	12	96
**PA-PhSO_3_H-PAN (0.1:1)**	1.62	1.57	2.17	4.13	0.45	0.96	0.72	1.6	1.28	1.91	18	6	84

**Table 3 molecules-27-04110-t003:** Temperature-dependent proton conductivities (S·cm^−1^) of the fabricated matrix-mixed membranes with different ratios at 98% RH.

Temperature	PA-PhSO_3_H-PAN (0.1:1)	PA-PhSO_3_H-PAN (0.4:1)	PA-PhSO_3_H-PAN (1:1)	PA-PhSO_3_H-PAN (3:1)
80 °C	7.61 × 10^−3^	1.18 × 10^−2^	2.89 × 10^−2^	4.90 × 10^−2^
70 °C	7.20 × 10^−3^	1.02 × 10^−2^	2.68 × 10^−2^	4.56 × 10^−2^
60 °C	6.73 × 10^−3^	9.54 × 10^−3^	2.35 × 10^−2^	4.13 × 10^−2^
50 °C	6.40 × 10^−3^	9.21 × 10^−3^	2.02 × 10^−2^	3.51 × 10^−2^
40 °C	5.90 × 10^−3^	8.86 × 10^−3^	1.82 × 10^−2^	2.99 × 10^−2^
30 °C	5.34 × 10^−3^	8.35 × 10^−3^	1.61 × 10^−2^	2.57 × 10^−2^

## Data Availability

Not applicable.

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
