# Peer review of "Highly Effective Proton-Conduction Matrix-Mixed Membrane Derived from an -SO3H Functionalized Polyamide"

_molecules, 2022, doi:10.3390/molecules27134110_

Round 1

Reviewer 1 Report

The manuscript titled Highly Effective Proton-Conduction Matrix-Mixed Membrane 2 Derived from a -SO3H Functionalized Polyamide was about the synthesis of composite proton exchange membranes from acyl chloride and amine precursors. The manuscript could be interesting to the readers of Molecules. However, some revisions and corrections are needed. And after reviewing the manuscript, comments are shown below:

1.     (Proton) ion conductivity depends on the membranes’ ion exchange capacity and water uptake. The IEC of the (composite and pristine) membranes should be determined, via NMR and/or titration, and reported.

2.     Line 156, what are built-in -SO3H moieties? Reference should be provided.

3.     Line 122, porosity not porocity

4.     The prepared membranes should be tested in a PEM fuel cell.

Author Response

Point-by-point response to reviewers’ comments

Reviewer 1:

Comment: The manuscript titled Highly Effective Proton-Conduction Matrix-Mixed Membrane Derived from a -SO3H Functionalized Polyamide was about the synthesis of composite proton exchange membranes from acyl chloride and amine precursors. The manuscript could be interesting to the readers of Molecules. However, some revisions and corrections are needed.

Answer: We express our sincere thanks to the reviewer for his/her valuable comments and recommendation for the publication after revisions. Please find our point-by-point responses below.

(1) Comment: “(Proton) ion conductivity depends on the membranes’ ion exchange capacity and water uptake. The IEC of the (composite and pristine) membranes should be determined, via NMR and/or titration, and reported.”

Answer: Many thanks for the reviewer’s valuable suggestion. We carried out the ion exchange capacity (IEC) measurements for all of the fabricated matrix-mixed membranes. The results were added in Table 2 and discussed in our revised manuscript. About the pristine PA-PhSO3H, it cannot be made into membrane directly, thus we only measured the ion exchange capacity of the fabricated matrix-mixed membranes. As described in our revised manuscript, the results were added in the sentences of “Physicochemical properties for our prepared membranes were then investigated according to the reported standard methods [43-46]. Firstly, an acid–base titration method was exploited to determine the ion exchange capacity (IEC) of fabricated matrix-mixed membranes [43,44]. After immersing the pre-weighted dried membranes in a 2 M NaCl solution for one day to replace the proton by sodium ions, the solution was titrated by 0.01 M NaOH solution (See details in experimental section in supporting information). The calculation results showed that the IEC of fabricated matrix-mixed membranes increases from 1.57% of PA-PhSO3H-MMM(0.1:1) to 2.89% of PA-PhSO3H-MMM(3:1), which agree well with their theoretical values as shown in Table 2”, lines 1-10, second paragraph of part 2.4. The experimental method was also described in the part of Experimental Section in the revised Supporting Information.

(2) Comment: “Line 156, what are built-in -SO3H moieties? Reference should be provided.”

Answer: Many thanks for the reviewer’s valuable comment. As described in the manuscript, we synthesized a -SO3H functionalized polyamide by one-pot acylation polymeric combination of acyl chloride and amine precursors (p-phenylenediamine sulfonic acid) and have named this material as PA-PhSO3H. As the PA-PhSO3H has in-built -SO3H group in its structure, which is derived from its reactant p-phenylenediamine sulfonic acid and no additional source of sulfonic acid is employed in its synthesis, so we called it as in-built -SO3H moiety. In our revised manuscript, we revised all of the phrases of “built-in” into “in-built”. Some publications were also used same phrase of “in-built” to describe the -SO3H functionalized materials such as in the ref. 27.

(3) Comment: “Line 122, porosity not porocity”

Answer: Many thanks for the reviewer’s valuable comment. In our revised manuscript, we revised the word of “porocity” into “porosity” in line 122.

(4) Comment: “The prepared membranes should be tested in a PEM fuel cell.”

Answer: Many thanks for the reviewer’s valuable suggestion. In our revised manuscript, we fabricated matrix-mixed membranes by the synthesized PA-PhSO3H and PAN, and measured their physicochemical properties, proton conductivity and long-life reuseability under various conditions. The results of these measurements reveals that the syntheized sulfonated polyamide is highly promising proton-conductive electrolyte and the fabricated matrix-mixed membranes are extremely efficient and robust PEM candidants used in PEM fuel cells, thus we will test them in a PEM fuel cell in the future works as we dicribed in the last second sentence of “As highly promising PEM candidates, we will test them in an integration PEM fuel cells in the future works” in part 2.4 in our revised manuscript.

Reviewer 2 Report

1. The authors synthesized the membranes based on polyamide and performed tests for proton conduction. However, the authors did not provide enough evidence for PEM materials. I think that the authors need to perform tests followed, at least;

a) theoretical and experimental IEC(ion exchange capacity);

b) WU (water uptake) and dimensional stability (wide and length);

c) chemical stability (fenton’s test).

 2. In Figure 3b, The PA-Ph shows low thermal stability than the PA-PhSO3H polymer. It is reasonable that the PA-PhSO3H starts decomposition at 200~250 oC because of the SO3H groups. However, PA-Ph polymer has to exhibit high thermal stability because of only aromatic with amide structure. Please check a reference [Progress in Polymer Science 35 (2010) 623–686] and explain the data why the PA-Ph is low stability. In addition, you need to explain the weight loss of both polymers up to 150 oC.

 3. In FT-IR, the sulfuric acid (ca. 1025) may be wrong. The OH group shows a broad peak at 3500~3200 cm-1 and acidic OH also appears in the same position. please indicate what the bending is clearly. In addition, PDSA didn't show the sulfuric acid group in Fig S2. Please check again.

 4. The author prepared the mixed membranes with different ratios of PAN in Fig S4 but measured the long-life reusability test of one of them (membrane : PAN=3:1) in Fig 5. Is there any reason to measure just one of the samples?

 5. In addition, the proton conductivity of PA-PhSO3H is in Table S1. But this study is the mixed membranes and the author demonstrates excellence in the proton conductivity of mixed one (3:1). Therefore, the proton conductivity of only PA-PhSO3H is mismatched in your study. you should exhibit the proton conductivities of all mixed membranes.

6. In figure S3, the author mentioned the amorphous powder was analyzed to observe the morphology of the membrane. I don’t understand how to confirm the morphology of the membrane using those images. The images were unclear to confirm the morphology of the separated hydrophilic and hydrophobic segments. please check those images again.

Author Response

Point-by-point response to reviewers’ comments

Reviewer 2:

(1) Comment: “The authors synthesized the membranes based on polyamide and performed tests for proton conduction. However, the authors did not provide enough evidence for PEM materials. I think that the authors need to perform tests followed, at least: a) theoretical and experimental IEC(ion exchange capacity); b) WU (water uptake) and dimensional stability (wide and length); c) chemical stability (fenton’s test).”

Answer: Many thanks for the reviewer’s valuable suggestions. We did all of the mentioned measurements for our fabricated matrix-mixed membranes and the results were added in our revised manuscript. The physicochemical properties of our fabricated matrix-mixed membranes were listed in Table 2 and discussed in part 2.4 as described in “Physicochemical properties for our prepared membranes were then investigated according to the reported standard methods [43-46]. Firstly, an acid–base titration method was exploited to determine the ion exchange capacity (IEC) of fabricated matrix-mixed membranes [43,44]. After immersing the pre-weighted dried membranes in a 2 M NaCl solution for one day to replace the proton by sodium ions, the solution was titrated by 0.01 M NaOH solution (See details in experimental section in supporting information). The calculation results showed that the IEC of fabricated matrix-mixed membranes increases from 1.57% of PA-PhSO3H-MMM(0.1:1) to 2.89% of PA-PhSO3H-MMM(3:1), which agree well with their theoretical values as shown in Table 2. Then, water uptake property and swelling characteristics as well as the dimensional stability were also tested for the fabricated matrix-mixed membranes (See details in experimental section in supporting information)[45,46]. After immersing the pre-weighted dried membranes overnight in distilled H2O at 30 and 80 °C, respectively, they were then rapidly dried by filter paper and weighted to calculate the water uptake. And the results showed that all the membranes show higher water uptake capabilities when treated them with higher temperature and the water uptake capabilities increase along with the sample ratio increase as shown in Table 2. With the same method, swelling characteristics and dimensional stability were also calculated. All of the fabricated matrix-mixed membranes show high dimensional stability with a swelling ratio less than 7% as shown in Table 2” in the second paragraph and “Subsequently, chemical and hydrolytic stabilities of the fabricated matrix-mixed membranes were also studied (See details in experimental section in supporting information). After immersing the prepared membranes in Fenton reagent (FeSO4 in H2O2 (3 %), 3.0 ppm) at 30 °C and 80 °C, respectively, the oxidative chemical stability was determined by the elapsed time (t) it took for the membrane to dissolve completely,[45] which shows all of the membranes have an elapsed time (t) of more than 6 h at 80 °C and even longer when tested them at 30 °C (Table 2). When they were immersed in distilled water at 50 °C, their elapsed time values until the membranes lost their mechanical properties were also recorded [44]. As shown in Table 2, all of the membranes show the elapsed time longer than 84 h, and the membrane of PA-PhSO3H-MMM(3:1) even shows an elapsed time upto 120 h. All these tests revealed the fabricated matrix-mixed membranes possess good physicochemical properties and should be high promising candidates of proton conductivity membranes applied in PEM fuel cells” in the third paragraph in part 2.4, the measurement methods were added in the part of Experimental Section in the revised Supporting Information and the related references were cited in Refs 43-46.

(2) Comment: “In Figure 3b, The PA-Ph shows low thermal stability than the PA-PhSO3H polymer. It is reasonable that the PA-PhSO3H starts decomposition at 200~250 oC because of the SO3H groups. However, PA-Ph polymer has to exhibit high thermal stability because of only aromatic with amide structure. Please check a reference [Progress in Polymer Science 35 (2010) 623–686] and explain the data why the PA-Ph is low stability. In addition, you need to explain the weight loss of both polymers up to 150 oC.”

Answer: Many thanks for the reviewer’s valuable comment. We did the TGA once again for both samples and redraw the Figure 3b. In our revised manuscript, we discussed the TGA results in the sentences of “The thermal stability of both samples was then determined using thermogravimetric analyses. When the temperature increased from room temperature upto 150 °C, weight losses were observed in both samples, which should be attributed to the loss of absorbed water molecules in their pores. [27] When the temperature was approximately 250 °C, significant weight losses due to polymer decomposition were observed in both samples (Figure 3b), indicating that they exhibit excellent thermal stability at the operating temperature of a PEM fuel cell (typically lower than 120 °C) [7-9]” in lines 3-10, part 2.2. We also cited the related publication in ref. 36.

(3) Comment: “In FT-IR, the sulfuric acid (ca. 1025) may be wrong. The OH group shows a broad peak at 3500~3200 cm-1 and acidic OH also appears in the same position. Please indicate what the bending is clearly. In addition, PDSA didn't show the sulfuric acid group in Fig S2. Please check again.”

Answer: Many thanks for the reviewer’s valuable comment. According to the references of [ACS Sustainable Chem. Eng. 2020, 8, 6, 2423–2432] and [Angew. Chem. Int. Ed. 2021, 60, 14875 –14880], the presence of -SO3H group in the range of 1000-1100 cm-1. That’s the reason we attribute the peak at 1025 cm-1 to -SO3H group in PA-PhSO3H that derived from the reactant of p-phenylenediamine sulfonic acid. In our revised manuscript, we cited the references in related descriptions and in ref. 21 and ref.27 and discussed in the description of “The strong absorption band at 1025 cm-1, which corresponds to the characteristic peak of -SO3H moieties [21, 27], was also identified in PA-PhSO3H, indicating that the sulfonated functional groups were also effectively integrated into the skeleton of synthesized PA-PhSO3H” in lines 9-13, part 2.1. Meanwhile, according to these references, the peaks at 1014 cm-1 in PDSA as shown in Figure S2 (a) should be the corresponding peak of the -SO3H group in the reactant PDSA.

(4) Comment: “The author prepared the mixed membranes with different ratios of PAN in Fig S4 but measured the long-life reusability test of one of them (membrane : PAN=3:1) in Fig 5. Is there any reason to measure just one of the samples?”

Answer: Many thanks for the reviewer’s valuable comment. We measured the long-life reusability of all the fabricated matrix-mixed membranes. In our revised manuscript, we discussed the results as shown in the description of “Additionally, the continuous test shows all of the fabricated membranes possess long-life reusability” the last third sentence of part 2.4 and we also showed the result in Figures S4-S6 in the revised Supporting Information.

(5) Comment: “In addition, the proton conductivity of PA-PhSO3H is in Table S1. But this study is the mixed membranes and the author demonstrates excellence in the proton conductivity of mixed one (3:1). Therefore, the proton conductivity of only PA-PhSO3H is mismatched in your study. You should exhibit the proton conductivities of all mixed membranes.”

Answer: Many thanks for the reviewer’s valuable suggestion. We measured the proton conductivities of all the fabricated matrix-mixed membranes. In our revised manuscript, we discussed the results as shown in the sentences of “The high proton conductivity of the synthesized PA-PhSO3H and the successful fabrication of robust matrix-mixed membranes based on PA-PhSO3H motivate us to evaluate the proton conductivity of the fabricated membranes. The EIS analyses results revealed that the proton conductivity increases along with the sample ratio increase in the membranes. It is only 7.61×10-3 S cm-1 for the membrane of PA-PhSO3H-MMM(0.1:1) at 353K and 98%RH, while it reaches up to 4.90×10-2 S cm-1 for the membrane of PA-PhSO3H-MMM(3:1) at the same condition (Table 3 and Figures S4-S6), the value of which is even comparable with those of commercial-available electrolytes being used in PEM fuel cells at the same temperature and relative humidity [18-20]” in lines 3-7, the last paragraph of part 2.4 and we also summarized the result in Table 3 in the revised manuscript and in Figures S4-S6 in the revised Supporting Information.

(6) Comment: “In figure S3, the author mentioned the amorphous powder was analyzed to observe the morphology of the membrane. I don’t understand how to confirm the morphology of the membrane using those images. The images were unclear to confirm the morphology of the separated hydrophilic and hydrophobic segments. Please check those images again.”

Answer: Many thanks for the reviewer’s valuable suggestion. We measured SEM of the synthesized PA-PhSO3H once again and the results were shown in the Figure S3. In our revised manuscript, we revised the description into “Scanning electron microscopy (SEM) images were then analyzed subsequently to observe the morphology of the synthesized PA-PhSO3H, and fine amorphous powder were observed from the resultant images as shown in Figure S3, which is agree very well with the result obtained from the PXRD measurements” in the lines 13-16, part 2.1.

Reviewer 3 Report

The method of proton conductivity using EIS is still arbitrarily a fitting of a simple circuit. It requires more refinement in terms of addressing the capacitive and resistive behaviors and other components of the circuit that could affect conductivity isolated -  like the Cu plates, RH environment, contact resistance?  

The highlights were mostly described focused on conductivity, can you comment on mechanical properties of these membranes/compounds? 

Can you show H+ conductivity, mechanical properties, swelling characteristics and gas/water permeation properties comparison against nation like or HSO3- based materials which are currently used in fuel cell research? 

Author Response

Reviewer 3:

(1) Comment: “The method of proton conductivity using EIS is still arbitrarily a fitting of a simple circuit. It requires more refinement in terms of addressing the capacitive and resistive behaviors and other components of the circuit that could affect conductivity isolated - like the Cu plates, RH environment, contact resistance?”

Answer: Many thanks for the reviewer’s valuable and professional comment. The proton conductivity can be regarded as a function of film drying/heating protocol, relative humidity, temperature and film thickness. We have measure AC impedance data and electrical properties of materials in a wide range of temperature (303-353K) and humid conditions (33%-98%) according to the reported method used as in the reference 27. In this method, home-made sample tablets and Cu plates are used as illustrated in the part of “Proton conductivity measurements of Pure Samples” in experimental section in our supporting information. For electrical measurements, the powder samples were firstly grinded homogeneously in mortars for 10 minutes and then cash-like discs a diameter of 6 mm and thickness ranging from 2 to 4.5 mm were prepared by compressed the materials under a pressure of 7 MPa for 30 s by a tablet press. Copper based electrodes and home-made sample holder were utilized, in which the prepared sample discs were placed for the measurement of resistivity and proton conductivity.

(2) Comment: “The highlights were mostly described focused on conductivity, can you comment on mechanical properties of these membranes/compounds?”

Answer: Many thanks for the reviewer’s constructive suggestion. We did more characterizations for our fabricated matrix-mixed membranes including ion exchange capacity (IEC), water uptake property, swelling ratios, dimensional stability, chemical and hydrolytic stability. These results were added in our revised manuscript. These properties of our fabricated matrix-mixed membranes were listed in Table 2 and discussed in part 2.4 as described in “Physicochemical properties for our prepared membranes were then investigated according to the reported standard methods [43-46]. Firstly, an acid–base titration method was exploited to determine the ion exchange capacity (IEC) of fabricated matrix-mixed membranes [43,44]. After immersing the pre-weighted dried membranes in a 2 M NaCl solution for one day to replace the proton by sodium ions, the solution was titrated by 0.01 M NaOH solution (See details in experimental section in supporting information). The calculation results showed that the IEC of fabricated matrix-mixed membranes increases from 1.57% of PA-PhSO3H-MMM(0.1:1) to 2.89% of PA-PhSO3H-MMM(3:1), which agree well with their theoretical values as shown in Table 2. Then, water uptake property and swelling characteristics as well as the dimensional stability were also tested for the fabricated matrix-mixed membranes (See details in experimental section in supporting information)[45,46]. After immersing the pre-weighted dried membranes overnight in distilled H2O at 30 and 80 °C, respectively, they were then rapidly dried by filter paper and weighted to calculate the water uptake. And the results showed that all the membranes show higher water uptake capabilities when treated them with higher temperature and the water uptake capabilities increase along with the sample ratio increase as shown in Table 2. With the same method, swelling characteristics and dimensional stability were also calculated. All of the fabricated matrix-mixed membranes show high dimensional stability with a swelling ratio less than 7% as shown in Table 2” in the second paragraph and “Subsequently, chemical and hydrolytic stabilities of the fabricated matrix-mixed membranes were also studied (See details in experimental section in supporting information). After immersing the prepared membranes in Fenton reagent (FeSO4 in H2O2 (3 %), 3.0 ppm) at 30 °C and 80 °C, respectively, the oxidative chemical stability was determined by the elapsed time (t) it took for the membrane to dissolve completely,[45] which shows all of the membranes have an elapsed time (t) of more than 6 h at 80 °C and even longer when tested them at 30 °C (Table 2). When they were immersed in distilled water at 50 °C, their elapsed time values until the membranes lost their mechanical properties were also recorded [44]. As shown in Table 2, all of the membranes show the elapsed time longer than 84 h, and the membrane of PA-PhSO3H-MMM(3:1) even shows an elapsed time upto 120 h. All these tests revealed the fabricated matrix-mixed membranes possess good physicochemical properties and should be high promising candidates of proton conductivity membranes applied in PEM fuel cells” in the third paragraph in part 2.4, the measurement methods were added in the part of Experimental Section in the revised Supporting Information and the related references were cited in Refs 43-46.

(3) Comment: “Can you show H+ conductivity, mechanical properties, swelling characteristics and gas/water permeation properties comparison against nation like or HSO3- based materials which are currently used in fuel cell research?”

Answer: Many thanks for the reviewer’s valuable comment. We did more characterizations for our fabricated matrix-mixed membranes including ion exchange capacity (IEC), water uptake property, swelling characteristics, dimensional stability, chemical and hydrolytic stability. These results were added in our revised manuscript. These properties of our fabricated matrix-mixed membranes were listed in Table 2 and discussed in part 2.4 as described in “Physicochemical properties for our prepared membranes were then investigated according to the reported standard methods [43-46]. Firstly, an acid–base titration method was exploited to determine the ion exchange capacity (IEC) of fabricated matrix-mixed membranes [43,44]. After immersing the pre-weighted dried membranes in a 2 M NaCl solution for one day to replace the proton by sodium ions, the solution was titrated by 0.01 M NaOH solution (See details in experimental section in supporting information). The calculation results showed that the IEC of fabricated matrix-mixed membranes increases from 1.57% of PA-PhSO3H-MMM(0.1:1) to 2.89% of PA-PhSO3H-MMM(3:1), which agree well with their theoretical values as shown in Table 2. Then, water uptake property and swelling characteristics as well as the dimensional stability were also tested for the fabricated matrix-mixed membranes (See details in experimental section in supporting information)[45,46]. After immersing the pre-weighted dried membranes overnight in distilled H2O at 30 and 80 °C, respectively, they were then rapidly dried by filter paper and weighted to calculate the water uptake. And the results showed that all the membranes show higher water uptake capabilities when treated them with higher temperature and the water uptake capabilities increase along with the sample ratio increase as shown in Table 2. With the same method, swelling characteristics and dimensional stability were also calculated. All of the fabricated matrix-mixed membranes show high dimensional stability with a swelling ratio less than 7% as shown in Table 2” in the second paragraph and “Subsequently, chemical and hydrolytic stabilities of the fabricated matrix-mixed membranes were also studied (See details in experimental section in supporting information). After immersing the prepared membranes in Fenton reagent (FeSO4 in H2O2 (3 %), 3.0 ppm) at 30 °C and 80 °C, respectively, the oxidative chemical stability was determined by the elapsed time (t) it took for the membrane to dissolve completely,[45] which shows all of the membranes have an elapsed time (t) of more than 6 h at 80 °C and even longer when tested them at 30 °C (Table 2). When they were immersed in distilled water at 50 °C, their elapsed time values until the membranes lost their mechanical properties were also recorded [44]. As shown in Table 2, all of the membranes show the elapsed time longer than 84 h, and the membrane of PA-PhSO3H-MMM(3:1) even shows an elapsed time upto 120 h. All these tests revealed the fabricated matrix-mixed membranes possess good physicochemical properties and should be high promising candidates of proton conductivity membranes applied in PEM fuel cells” in the third paragraph in part 2.4, the measurement methods were added in the part of Experimental Section in the revised Supporting Information and the related references were cited in Refs 43-46. The results of these measurements reveals that the syntheized sulfonated polyamide is highly promising proton-conductive electrolyte and the fabricated matrix-mixed membranes are extremely efficient and robust PEM candidants used in PEM fuel cells, thus we will test them in a PEM fuel cell in the future works as we dicribed in the last second sentence of “As highly promising PEM candidates, we will test them in an integration PEM fuel cells in the future works” in part 2.4 in our revised manuscript.

Round 2

Reviewer 2 Report

IEC value has no unit. please recheck it.

The PA-PhSO3H-MMM should be changed to the name " PA-PhSO3H-PAN" for other researchers.

Reviewer 3 Report

Looks clear and all questions were addressed. I would still refine the proton conductivity measurement techniques using alternate methods as well as using EIS. Looking forward for a follow up publication to testing of these membranes in a fuel cell experiments and catalyst synthesis studies with these ionomers/powders. I would also recommend comparison against Nation membranes with similar thickness to study performance testing.